# Multidisciplinary Approach for Hypothalamic Obesity in Children and Adolescents: A Preliminary Study

**DOI:** 10.3390/children8070531

**Published:** 2021-06-22

**Authors:** Daniele Tessaris, Patrizia Matarazzo, Gerdi Tuli, Antonella Tuscano, Ivana Rabbone, Alessandra Spinardi, Antonella Lezo, Giorgia Fenocchio, Raffaele Buganza, Luisa de Sanctis

**Affiliations:** 1Pediatric Endocrinology Unit, Regina Margherita Children’s Hospital, University of Turin, 10126 Turin, Italy; pmatarazzo@cittadellasalute.to.it (P.M.); gerdi.tuli@unito.it (G.T.); ivana.rabbone@unito.it (I.R.); raffaele.buganza@unito.it (R.B.); luisa.desanctis@unito.it (L.d.S.); 2Postgraduation School of Pediatrics, University of Turin, 10126 Turin, Italy; antonella.tuscano@unito.it; 3SUISM, University of Turin, 10126 Turin, Italy; alessandra.spinardi@libero.it; 4Dietetic and Clinical Nutrition Unit, Regina Margherita Children’s Hospital, 10126 Turin, Italy; antonella.lezo@unito.it; 5Clinical Psychology, Regina Margherita Children’s Hospital, 10126 Turin, Italy; g.fenocchio@gmail.com

**Keywords:** hypothalamic obesity, weight gain, BMI, metformin

## Abstract

Hypothalamic obesity (HO) is delineated by an inexorable weight gain in subjects with hypothalamic disorder (congenital or acquired). The aim of the present study was to evaluate the effect of a multidisciplinary approach on weight trend and metabolic outcome in children and adolescents with hypothalamic disease who were overweight or obese. Thirteen patients (aged 8.1–16.1 years) received a personalized diet, accelerometer-based activity monitoring, and psychological assessment. Height, weight, body mass index (BMI), and serum metabolic parameters were assessed at baseline (T0) and after six months (T1). Metformin was introduced at T1 in four subjects who were then re-evaluated after six months (T2). At T1, weight gain was significantly reduced compared with T0 (0.29 ± 0.79 kg/month vs. 0.84 ± 0.55 kg/month, *p* = 0.03), and weight standard deviation score (SDS) and BMI SDS did not change significantly, as serum metabolic parameters. The four subjects treated with metformin showed a reduction of weight SDS and BMI SDS at T2. In conclusion, patients treated with our multidisciplinary approach showed, after 6 months, favorable results characterized by decreased weight gain and stabilization of weight SDS and BMI SDS in a condition usually characterized by inexorable weight gain. However, further analysis, larger cohorts, and longer follow-up are needed to confirm these preliminary data.

## 1. Introduction

Hypothalamic obesity (HO) is a serious and heterogeneous disease related to several conditions, such as hypothalamic damage caused by tumors (e.g., craniopharyngioma, glioma, germinoma, hamartoma) and/or their treatment, inflammatory diseases (e.g., sarcoidosis, tuberculosis, histiocytosis, encephalitis), trauma, neurosurgery, cerebral aneurysm, single-gene mutations (e.g., leptin, leptin receptor, CART, POMC), rare congenital disorders with midline malformations (e.g., septo-optic dysplasia, mutations of the LHX3 gene), genetic syndromes (e.g., Prader Willi syndrome, Bardet-Biedl syndrome), and the use of psychotropic drugs (e.g., antidepressants, mood stabilizers, antipsychotics) [1,2,3].

The presence of HO may be considered if the following criterion is met: evidence of a pathologic process injuring the hypothalamus with obesity that develops subsequently to or in association with that injury [4]. In general, the weight gain pattern in HO patients is characterized by sudden onset and rapid acceleration after hypothalamic damage [4].

The rapid and continuous weight gain in HO is the result of multifactorial pathogenetic mechanisms [4,5,6,7,8]. The hypothalamus regulates body weight by balancing food intake, energy expenditure, and body energy stores; it receives afferent messages from adipose tissue, the gastrointestinal tract, liver, and pancreatic beta cells, and sends efferent messages to the same organs, as well as to the muscles and is linked through direct synaptic connections to appetite-regulating regions of the brain, such as the limbic system [1]. Several nuclei in the medial hypothalamus are key regulators of satiety and energy expenditure. The nuclei that seem most commonly implicated in HO are the arcuate nucleus, the paraventricular nucleus, the ventromedial nucleus, the dorsomedial nucleus, and the dorsal hypothalamic area [7]. HO may also be accompanied by a series of hormonal alterations due to imbalance of hypothalamic-pituitary function, such as a reduced secretion of growth hormone, gonadotropin, and thyroid-stimulating hormone and/or an impaired circadian rhythm that exerts a huge influence on the metabolic state. Furthermore, reduced physical activity related to possible coexistent disabilities (e.g., visual and neuromotor impairment) significantly decreases energy expenditure. Finally, a low-resting metabolic rate and autonomic imbalance are also common findings in this condition [9,10,11,12].

Abdominal obesity, associated with dyslipidemia, hyperinsulinemia caused by insulin resistance, and elevated blood pressure, which are all known markers of the metabolic syndrome in adulthood, may therefore be present in these patients and contribute to an increased risk of cardiovascular disorders and long-term mortality [13,14,15].

Patients with HO are usually refractory to lifestyle changes programs aimed at caloric restriction and increased physical activity [8]. Extreme hyperphagia has been noted as a hallmark of HO, but obesity can develop even in the absence of hyperphagia [4].

There is no standard pharmacological intervention that has consistently been shown to have a positive effect in these complicated-to-manage patients. Due to the rarity of HO and the objective difficulties in testing new drugs in the pediatric age, few reports or clinical studies on its treatment in children and adolescents have been reported so far. These studies are small and limited by significant side effects or have shown only limited or no effects. Most show that HO is often intractable and mostly resistant to classical approaches based on diet and physical activity [8].

Regarding hyperinsulinism, typically present in HO, treatment with octreotide was initially suggested [16], but this has shown significant side effects (particularly diarrhea and abdominal discomfort) and unclear benefits [7]. Combined therapy of diazoxide and metformin has also been proposed, with reduction of weight gain and BMI [17], but this study had limitations due to the small sample size and the significant adverse events encountered (gastrointestinal symptoms, peripheral edema, elevated liver enzymes). The combination of metformin and fenofibrate in one study did not improve weight or BMI [18]. Considering the energy expenditure, tri-iodothyronine (T3) has been tested [19] but even this therapeutic option does not appear to be effective [20]. Based on the observation that patients with HO have impaired sympathoadrenal activation and epinephrine production, it has been suggested that this disorder may be treated with amphetamine derivatives and CNS stimulants, particularly with dexamphetamine [21,22,23], as well as caffeine and ephedrine [24]; such drugs have been shown to have a positive impact on weight gain, BMI, physical activity, and daytime wakefulness in a number of cases, although all of these treatments need further studies to be validated. Studies on the administration of glucagon-like peptide-1 receptor agonist (GLP-1 RA) have shown varying degrees of success [25,26], while methionine aminopeptidase 2 [27] was successful in lowering body weight, but dosing was discontinued early due to venous thrombotic events. Recently, oxytocin has shown potential benefits for the treatment of obesity and has been proposed for HO [28], also in combination with naltrexone [29]. Bariatric surgery in adolescents with primary obesity is usually considered a last resort and is rarely performed. HO is a rare form of obesity and, although there is the possibility of considering pediatric bariatric surgery as a therapeutic option for adolescents [1], surgical experience in pediatric HO is virtually non-existent, the long-term effects can be questioned, and serious complications can occur after surgery, so its risk-benefit ratio must be considered with great caution [8]. In a recent long-term outcome study in severe early-onset obesity due to genetic mutations in the leptin-melanocortin signaling cascade, patients initially experienced weight loss after bariatric surgery, but this was followed by substantial weight regain [30].

The aim of our study was to evaluate the efficacy of a multidisciplinary approach, which includes endocrine care of affected subjects together with a dietary intervention, a physical exercise program, a psychological assessment with standardized scales, and, in a selected cohort of patients, a drug treatment with metformin (which was proposed by several authors for primary obesity [31]).

Although HO is often frustratingly refractory to lifestyle interventions, a healthy diet should always be encouraged [4]. The role of exercise (particularly in cases without neuromotor impairment) is also crucial and can help counteract the impaired sympatho-adrenal activation in HO [4]; in the study, we introduced an accelerometer-based activity monitor, previously utilized to estimate the physical activity of HO subjects [10], not only to assess the amount of daily movement, but also with a motivation role for patients and their family, giving them a tool to quantify the results of their day-to-day commitment.

There are few studies on lifestyle and dietary intervention in pediatric HO [8] and many of them have not reported any improvement in weight trend. Considering the absence of a pharmacological treatment of clear and proven efficacy, in this preliminary study we proposed a multidisciplinary approach with a simple and defined therapeutic pathway, defining the role of the different specialists, to help clinicians in the follow up of these patients.

## 2. Materials and Methods

### 2.1. Inclusion Criteria

Thirteen children and adolescents with hypothalamic disease, 5 males and 8 females aged between 8.1 to 16.1 years (mean 12.2), were enrolled at the Pediatric Endocrinology Unit of Regina Margherita Children’s Hospital in Turin (Italy), where they were already on endocrinological follow-up (mean follow-up of 6.7 years for patients with tumors of the hypothalamic-pituitary area and from birth for patients with congenital pathology). In all subjects, brain MRI evidenced hypothalamic damage.

The inclusion criteria were age >8 years at the start of the study and overweight or obese, with weight and BMI worsening with standard care. Overweight and obese were defined according to the BMI standards proposed by Cole et al. [32], which are based on and linked to the corresponding adult BMI cut-offs (25 and 30 kg/m^2^ for overweight and obese, respectively).

### 2.2. Study Design

The study included four consecutive steps.

First step: start of the multidisciplinary treatment (time-point T0).

The study involved different specialists: a pediatric endocrinologist, a pediatric diabetologist with experience in the management of obesity, a dietician, and a psychologist and graduate in motor and sport sciences.

All of the visits and exams were performed in a single day and in the presence of the parents.

At initial evaluation, weight gain in the 6 months prior to enrollment, height, weight, and BMI were recorded. Height, weight, and BMI were also reported as standard deviation score (SDS) for age and sex. Height was measured to the nearest 0.1 cm with a calibrated stadiometer and body weight was evaluated to the nearest 0.1 kg using a calibrated scale.

Blood exams were performed to evaluate serum glucose, insulin, HOMA (Homeostatic Model Assessment) index [33], triglycerides, total cholesterol, and HDL cholesterol, in addition to routine hormone tests to assess the adequacy of ongoing replacement therapy.

The pediatric endocrinologist evaluated the patients and hormone tests and established hormone replacement therapy.

The pediatric diabetologist with experience in the management of obesity examined the patients and evaluated the metabolic analyses, as well as the presence of any additional complications of obesity.

The dietary assessment, aimed to investigate micronutrients and macronutrients intake, was performed combining two methods: the 24-h dietary recall, a structured interview intended to capture detailed information on all foods and beverages consumed by the respondent in the past 24 h, and the 3-day dietary records, with self-reported food diaries (the latter method provides more data, but has limitations mainly due to the tendency of subjects to report food consumption close to those which are socially desirable). The dietician evaluated any differences in terms of caloric intake obtained from the two different methods and developed a personalized diet for each patient, considering the nutritional history and the caloric requirement based on resting energy expenditure and physical activity.

A psychological assessment was performed by the psychologist using Vineland Adaptive Behavior Scales [34], a standardized assessment tool which utilizes a semi-structured interview to evaluate individuals from birth to adulthood in their functional personal and social abilities and produces standard scores in four domains, i.e., communication, daily living skills, socialization, and motor skills. In addition to age-equivalent scores for the raw domain scores, the measure also yields an overall Adaptive Behavior Composite Standard score. It allows for the identification of the strengths and weaknesses of the subject in specific areas of adaptive behavior, favoring the planning of individual educational or rehabilitative interventions.

To plan physical activity, the history of motor activity was evaluated, also considering the visual or neuromotor deficit of some patients.

At the end of this first phase, the multidisciplinary team provided the families with a final report including the result of the visits, analyses, and dietary and lifestyle indications, and provided them with an accelerometer-based activity tool (MyWellness Key, Technogym, Cesena, Italy) for daily monitoring of physical exercise (number of movements per day with a goal of 500 movements). It is a portable device with an accelerometer that can be worn by the patient and can be used to monitor physical activity, measuring the duration and intensity of movement during normal daily life. The purpose of the device, in addition to monitoring the patient’s daily physical activity, was motivational; every time the child reached the predetermined number of movements, a “+” sign appeared on the device screen, which indicated the good work done, motivating them to increase the activity.

After one month, a second multidisciplinary visit was carried out with patients and parents to reinforce the lifestyle change, motivate the patients in the continuation of their programs, highlight any difficulties in relation to the diet, and evaluate the first data and the correct use and functioning of the accelerometer-based activity monitor.

Second step: endpoint assessment (time-point T1).

Therapeutic effectiveness was evaluated 6 months after T0 (time-point T1), with another multidisciplinary control.

We defined the achievement of at least one of following criteria as a response to treatment: reduction of weight SDS for age and sex [35]reduction of BMI SDS for age and sex [35]reduction in weight gain compared to the previous 6 monthsimprovement of the HOMA index.

Third step: pharmacological treatment with Metformin at T1 in patients aged >10 years old plus at least one of the following criteria: severe/morbid obesity (according to Cole’s standards, based on BMI 35 at age 18 [32]);serum insulin levels >30 mUI/LHOMA index >4weight gain after six months of multidisciplinary treatment plus HOMA index >3

To minimize gastrointestinal side effects, metformin was started at a dose of 250 mg twice daily over 10 days, then increased to 500 mg twice daily.

Fourth step: endpoint assessment (time-point T2).

Therapeutic efficacy of metformin, added to lifestyle changes, was evaluated after six months and included the same criteria for time-point T1.

The study was conducted according to the guidelines of the Declaration of Helsinki, and approved by the Ethics Committee of Regina Margherita Children Hospital (protocol number 0072684). Informed consent was obtained from parents and, when possible (considering visual and neuromotor impairment in some cases), from patients involved in the study.

### 2.3. Statistical Analysis

Statistical analysis was performed by SPSS V.27. Continuous variables were expressed as mean and SDS. The paired two-tailed *t*-test was used to compare variables measured in the same subject at different time-points, while the independent *t*-test was used for statistical analysis between different groups of subjects at the same time-point.

Partial correlation was used to explore the relationship between perceived control of internal states (measured by the Perceived Stress scale), controlling for sex and age. Preliminary analyses were performed to ensure no violation of the assumptions of normality, linearity, and homoscedasticity.

## 3. Results

The study included eight patients with parasellar tumors (five craniopharyngiomas, three hypothalamic astrocytomas) and five patients with congenital midline malformations (four septo- optic dysplasia, one LHX3 mutation).

All subjects presented with endocrine dysfunction. In particular, 12 had multiple pituitary deficiency requiring hormone replacement therapy, 1 subject had multiple hypopituitarism and precocious puberty (patient 7), and 1 subject (patient 8) had precocious puberty and syndrome of inappropriate antidiuresis (SIAD).

All patients were taking an appropriate dose of hormone replacement treatment with good hormonal control during the course of the study.

The main clinical and auxological data at T0 are reported in Table 1.

At T0, nine patients were obese and four were overweight according to Cole’s standard [32].

All 13 subjects received the personalized diet and psychological assessment.

The average caloric intake of the diets administered was 1525 kcal/day, ranging from 1300 to 1700 kcal/day; the average caloric intake per kg was 27.0 kcal/kg, ranging from 45.9 kcal/kg to 12.8 kcal/kg (Table 2).

Of the subjects, 10 out of 13 regularly utilized the accelerometer-based activity tool for physical exercise monitoring (three patients reported technical problems with the device).

Analyzing daily physical exercise (number of movements per day) through the accelerometer-based activity tool from T0 to T1, 50% of patients reached the goal of an average of 500 movements per day (<500 was considered “low active”). The average of the corresponding calculated daily energy consumption (which is related to the number of daily movements performed and is influenced by the subject’s body weight) was 196.4 kcal/day, ranging from 100 kcal/day to 322 kcal/day (Table 2).

Psychological assessment with Vineland Adaptive Behavior Scales showed an equivalent age lower below the mean age (12.2 years) in our population for all the abilities. Specifically, it indicated an average age of 3.7 years for motor skills, 7.6 years for daily skills (including eating, dressing, personal hygiene, housework, etc.), and 8.4 and 8.5 years for socialization and communication, respectively (Figure 1).

All 13 patients completed the assessment at T1.

The main clinical and laboratory results at T0 and T1 are shown in Table 3.

At T1, the main effect was observed for weight gain, which was significantly reduced compared with T0 (0.29 ± 0.79 kg/month vs. 0.84 ± 0.55 kg/month, *p* = 0.03) (Figure 2). In particular, weight gain decreased in 9/13 subjects from T0 to T1.

At T1, weight SDS decreased in 9/13 subjects and BMI SDS in 8/13 compared with T0, but the overall result did not show a statistically significant difference from T0 to T1 for these parameters.

The analyses were then stratified by disease (tumors vs. congenital disease), age (<12 vs. ≥12 years, which represents the average age of the subjects examined), and sex (male or female).

Males showed a greater reduction of BMI from T0 to T1 than females (−0.24 ± 0.29 vs. +0.01 ± 0.11, *p* = 0.04), while there were no significant differences when comparing patients with tumors with those with congenital diseases, nor by stratifying the subjects by age; a better response was recorded in children <12 years, but without statistical significance.

Considering the greater reduction of BMI in males and since the participants age range was very large, we evaluated the possible effect of sex and age on the overall results. When we compared the change in the Pearson R correlation coefficient from zero order (BMI T0 and T1 not controlled: r = 0.982, *p* < 0.001) to whole model (controlled for sex and age: r = 0.989, *p* < 0.001), we found a very low effect of sex and age on the relationship of BMI values and we can conclude that they have no significant effect on two-times repeated BMI measurements. Analogues results were obtained for weight (zero order correlation r = 0.977, *p* < 0.01; for sex and age r = 0.978, *p* < 0.001) and weight gain (zero order correlation r = 0.350, *p* = 0.241; whole model corrected for sex and age r = 0.304, *p* = 0.363). The same analyses led to the same results considering sex and age separately on BMI, weight, and weight gain.

Serum insulin levels, HOMA index, triglycerides, total cholesterol, and HDL cholesterol did not change significantly from T0 to T1 (Table 4).

Four patients proceeded to T2, starting metformin treatment (250 mg twice daily over 10 days, then increased to 500 mg twice daily) at T1, according to the aforementioned criteria. Patients and their parents reported regular assumption of therapy.

These patients showed reduction of weight, weight SDS, and BMI SDS from T1 to T2 (Table 4).

As regards the HOMA index, the reduction was more relevant from T0 to T1 than from T1 to T2; after the introduction of metformin, 3/4 subjects showed a slight further reduction. The other metabolic parameters (insulin, triglycerides, total cholesterol, HDL cholesterol) did not show a clear trend.

The limited number of subjects treated with metformin did not allow for statistical analysis. No liver or gastrointestinal dysfunction or other side effects were reported during treatment with metformin.

## 4. Discussion

Children and adolescents with hypothalamic diseases often develop severe organic obesity, which can worsen their metabolic profile and lead to increased morbidity and mortality, resulting in reduced quality of life. Clinical management of HO is extremely difficult due to lack of response to lifestyle modification and the absence of clearly effective treatments. Furthermore, the experiences in the treatment of HO in the pediatric population are nowadays particularly limited [7].

Among the causes of HO in the literature, as in our study, the most extensively documented in children and young adults is that linked to craniopharyngioma, from the tumor itself or as a sequela of treatment. Craniopharyngioma is a nonglial intracranial tumor of the sellar and/or parasellar region that represents 1.2–4.0% of all childhood intracranial tumors [37]. Survival rates range from 91 to 98%, but the quality of survival is often compromised due to proximity to optical, pituitary, and hypothalamic structures, resulting in loss of vision and endocrinopathies. Most (85–95%) of patients with craniopharyngioma suffer from multiple hypothalamic-pituitary deficiencies, as well as cerebrovascular problems and neurologic and neurocognitive dysfunction. Obesity and eating disorders, correlated with the degree and extent of hypothalamic damage, are observed in 40–50% of children with craniopharyngioma, resulting in reduced energy expenditure and increased risks of metabolic syndrome, cardiovascular disease including sudden death events, multisystem morbidity, and even mortality in these patients [37].

Our study also included cases of HO following astrocytoma, the most frequent cerebral tumor in the pediatric age, occurring in subjects with septo-optic dysplasia, a rare condition defined by variable combination of dysgenesis of midline brain structures including optic nerve hypoplasia and hypothalamic-pituitary dysfunction often associated with a wide variety of cerebral malformations of cortical development, and a child with causative mutation in the LHX3 gene.

The usual cornerstone of obesity treatment is a lifestyle changes program aimed at caloric restriction and increased physical activity and, although HO can seem frustratingly refractory to such measures, as confirmed by the few studies on lifestyle and dietary intervention in pediatric HO [8], a healthy diet and exercise must absolutely be encouraged [4], also considering the absence of pharmacologic treatments with evident and proven benefits.

We described our multidisciplinary approach, defining the role of the different specialists of the multidisciplinary team and proposing the use of a qualitative/quantitative psychological assessment with Vineland Adaptive Behavior Scales and of an accelerometer-based activity monitor to reinforce the motivation for daily activity.

Disabilities and motor impairments, also identified by the psychological evaluation with Vineland Adaptive Behavior Scales in our study, can reduce physical activity and adherence to the clinical indications. The application of this scale can improve the knowledge of specific patient limits or strengths to better plan approaches and directions, including physical activity.

The main goals of the study were to reduce weight gain, weight SDS, or BMI SDS and to counteract metabolic derangement, particularly hyperinsulinemia. Our approach aimed at improving the adherence of patients and their families to the multidisciplinary therapeutic strategies and at implementing the collaboration of different medical professionals. In pediatric HO, environmental factors are more controlled by adult family members and the family context plays a fundamental role in the process of treating childhood obesity.

Our approach has shown that organized, low-intensity care without frequent hospitalizations, but with periodic outpatient checkups and some additional resources can achieve significant results, particularly by reducing weight gain without further weight gain SDS and BMI SDS in a condition characterized by relentless weight gain, as shown in several studies [8].

We obtained a statistically significant reduction in weight gain over a 6-month follow-up period, with substantial stabilization of weight SDS and BMI SDS; at T1, compared with T0, weight SDS and BMI SDS decreased in most of the subjects even if with overall results were not statistically significant.

The reduction in weight gain was a certification of patients and family members adherence to the proposed program; at the same time, the partial results of physical exercise and an equivalent lower average age in all skills reflect the presence of important physical and intellectual disabilities that can reduce the possibility of practicing physical activity, which is a major factor accounting for obesity in HO patients [10] and requires significant motivation from patients and families, which we have tried to increase with the use of a digital device.

Considering the etiology of the hypothalamic damage and the age and sex of the patients, we did not find substantial differences between subjects with congenital or acquired diseases, while we found a better response in younger males, a category that itself may be more interested in physical activity changes and in the use of technological devices. In general, our result highlighted the importance of early combined prevention and assessment of obesity in pediatric patients with hypothalamic disorders, but considering the small number of subjects in the different subpopulations, we cannot obtain certain answers nor find a clear explication on the different responses in males compared to females.

In four patients with more severe conditions or without improvement after the 6-month non-pharmacological multidisciplinary approach, metformin was introduced, yielding a reduction in weight, weight SDS, and BMI SDS and no clear effects on HOMA and other metabolic indexes. Given the small number of subjects, we cannot draw conclusions on the effect of metformin, but with this preliminary study we propose its use in selected cases to evaluate its effectiveness on a larger case series.

The rationale for the use of metformin in HO is supported by the role of hyperinsulinemia in the inhibition of the sympathetic nervous system, beside the classical insulin resistance arising from the worsening obesity. Recent data indicate that a proper balance of the autonomic nervous system is crucial for metabolism; it is well known that adipose tissue is richly innervated by sympathetic nerve fibers that control lipolysis, and it now appears that lipogenesis is also controlled by parasympathetic innervation of adipose tissue originating from separate sympathetic and parasympathetic neurons in the periventricular nucleus and suprachiasmatic nucleus [37,38]. The relationship between hyperinsulinism and HO was deeply investigated, and it was postulated that ventromedial nucleus (VMH) lesions lead to autonomic imbalance from disinhibited or increased vagal tone with hyperinsulinemia as the result of overactive vagal transmission, and that these events are major contributors to the pathogenesis of HO. Supporting these theories, studies on rats demonstrated that rodents with VMH lesions developed signs of decreased sympathetic nervous system function and hyperinsulinemia even in the absence of hyperphagia or obesity, and pancreatic vagotomy prevented the exaggerated secretion of glucose-stimulated insulin in VMH lesioned rats; furthermore, higher-fasting insulin levels were detected among patients with HO compared with individuals with a common diet [4,39,40,41].

In HO, treatment with octreotide, a somatostatin-5 receptor agonist on the beta-cell, which is coupled to and inhibits the voltage-gated calcium channel, has shown discordant results and its use is limited by the frequent abdominal side effect [7]. The use of metformin has been proposed in combination with diazoxide [17], in a study with a small sample size and notable adverse events, and with fenofibrate, without the improvement of weight or BMI in 22 subjects [18].

In patients with primary obesity, metformin is frequently used to reduce weight, BMI, and insulin resistance and to counteract the complications associated with obesity, such as type-2 diabetes mellitus, metabolic syndrome, cardiovascular diseases, and liver diseases. It has been associated with weight loss, possibly due to a combination of its inhibition of gluconeogenesis, increased peripheral glucose utilization, increased fat oxidation in skeletal muscles, and inhibition of lipid synthesis in the liver, though its pharmacokinetics are not completely understood [42]. A recent meta-analysis [31] evaluating several studies in children and adults found a greater reduction in BMI in overweight or obese patients taking metformin than in controls in more than half the studies, and progression toward T2DM was significantly reduced in adults using metformin. The effects of metformin on weight/BMI varied, with smaller reductions in children than in adults, and the authors supposed that this could be related to differences in adherence, daily dosage, and insulin status.

In our study, the limited number of subjects treated with metformin did not allow for definitive conclusions on the effects of metformin on weight trend; two patients improved in terms of weight and BMI from T0 to T1, while all four patients improved the same two parameters together with insulin resistance with the addition of metformin from T1 to T2. Regarding the assessment of insulin resistance, the HOMA index was used because it compares favorably with other models and has the advantage of requiring a single plasma sample for insulin and glucose, while the gold-standard glucose clamp technique is far more invasive; however, it should be considered that the HOMA index has some limitations, since it is poorly reproducible, insulin secretion is pulsatile, and cut-off values differ by races, ages, and sexes, for which several different parameters were proposed to evaluate insulin resistance [43,44,45].

Our pilot study had different limitations: small sample, large age range, lack of a control group and short follow-up, and potential interference of other hormone treatments being taken by the participants.

Therefore, further standardized clinical studies in the pediatric age are needed to indicate the real efficacy of this approach in the HO population, in which clinicians are faced with considerable hormonal, metabolic, and care complexity.

In absence of an effective pharmacological treatment, a multidisciplinary lifestyle intervention remains the initial cornerstone of HO treatment, reconsidering the objectives in the light of progress, encouraging physical activity, and adherence to dietary advice and evaluating the specific characteristics of the individual over time.

## 5. Conclusions

We proposed a multidisciplinary approach with low intensity care which, in our study, led to the reduction of weight gain after 6 months in a condition usually considered inexorable in terms of gaining body weight and without pharmacologically effective treatment. This pilot study provided preliminary data which support the utility of studies with larger populations and longer follow-ups based on this approach. We also introduced the use of a qualitative/quantitative psychological assessment with Vineland Adaptive Behavior Scales to better assess patient limitation and strength and a device to monitor and reinforce motivation in daily activity and proposed the use of metformin treatment in selected cases.

The experiences in the treatment of HO in the pediatric population are, nowadays, particularly limited and we believe this experience, further ongoing to gather longer-term follow-up data in a larger cohort, will provide evidence that, at the time of diagnosis, a multidisciplinary, personalized approach should be started and then continued throughout life to prevent, over time, relentless weight gain and the complications and deterioration of the quality of life of which this extremely disabling condition is characterized.

## Figures and Tables

**Figure 1 children-08-00531-f001:**
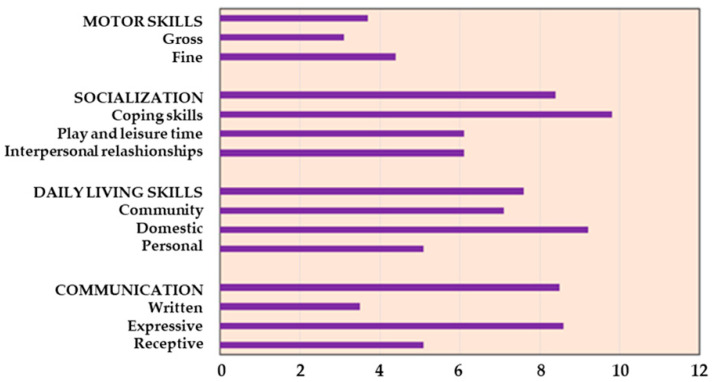
Evaluation of equivalent age according to Vineland Adaptive Behavior Scales.

**Figure 2 children-08-00531-f002:**
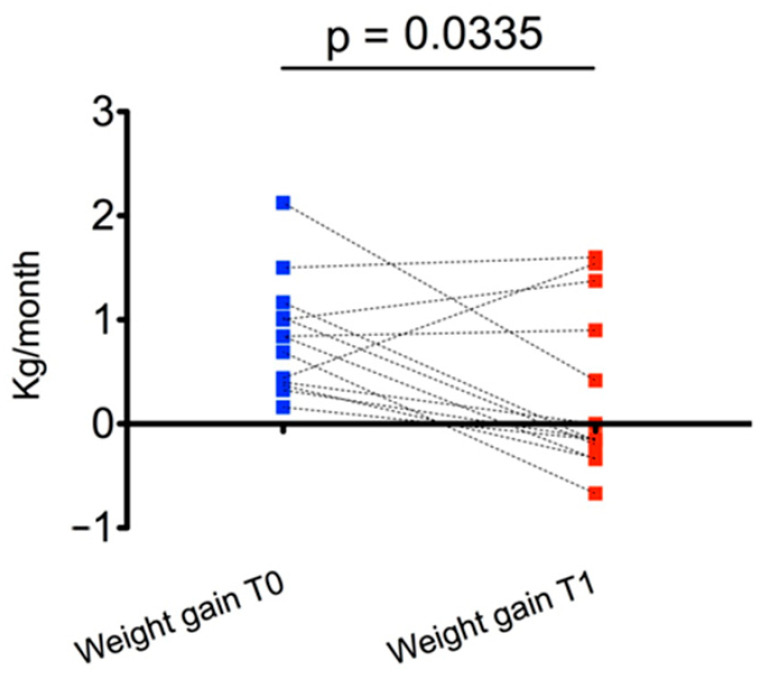
Changes in weight gain (kg/month) during the six months before and the six months after multidisciplinary management.

**Table 1 children-08-00531-t001:** Main baseline clinical and auxological data of the 13 enrolled subjects.

Subject	Sex	Disease *	Treatment **	Age (Years)	Height (cm)	Height (SDS)	Weight (kg)	Weight (SDS)	BMI (kg/m²)	BMI (SDS)
1	F	C	H-L-G-E-D	16.1	171.7	1.38	113	4.16	38.3	3.43
2	F	C	H-L-G-E-D	15.1	154.2	−1.31	57	0.42	23.9	1.27
3	F	C	H-L-G-E-D	13.8	166	1.10	96	3.72	34.8	3.23
4	F	C	H-L-G-E-D	13.3	143.5	−1.98	52.5	0.54	25.5	1.87
5	F	C	H-L-G-E-D	10.3	145.5	0.78	62.5	2.77	29.5	3.08
6	F	A	H-L-E	14.7	153.5	−1.28	97	3.71	41.1	3.76
7	M	A	H-L-D-Tr	8.8	113.4	−3.28	30.5	0.54	23.7	2.72
8	F	A	Tr-To	8.1	126.4	−0.30	44	2.62	27.5	4.76
9	M	SOD	H-L-G-D	12.4	148.5	−0.28	84	3.18	38.3	3.62
10	M	SOD	H-L-G	9.3	120.6	−2.40	35	1.02	24.3	3.88
11	M	SOD	H-L-G-Te	12.9	141.7	−1.57	46.8	0.48	23.3	1.78
12	F	SOD	H-L-G-D	13.6	155.5	−0.35	82	2.95	33.9	5.15
13	M	LHX3	H-L-G-Te	12.3	150	−0.01	53	1.43	23.56	1.95

* C = craniopharyngioma; A = astrocytoma; SOD = septo-optic dysplasia; LHX3 = LHX3 mutation. ** H = hydrocortisone; L = levothyroxine; G = rhGH; E = estroprogestin; D = desmopressin; Tr = triptorelin; To = vaptan; Te = testosterone. Patients 6 and 7 had GH deficiency but they were not on rhGH treatment due to their residual tumors.

**Table 2 children-08-00531-t002:** Caloric intake of the diets administered, resting energy expenditure, and data from the accelerometer-based activity monitor.

Subject	CI (kcal/Die)	BEE (kcal/Day)	Mov/Day	DEC (kcal/Die)
1	1450	1851	-	-
2	1400	1487	512.5	237.5
3	1300	1746	548.8	322
4	1500	1278	530	244.4
5	1500	1313	-	-
6	1600	1478	228.5	133
7	1400	956	407.8	108.2
8	1500	1081	600	172.4
9	1700	1349	-	-
10	1400	1024	495.3	169.1
11	1600	1251	353.3	100
12	1700	1513	655.4	314.6
13	1700	1384	442.6	162.9

CI = caloric intake; BEE = basal energy expenditure calculated with Schofield formula [36]; Mov = movements; DEC = daily energy consumption due to movements detected by the accelerometer.

**Table 3 children-08-00531-t003:** Clinical and metabolic parameters at T0 and T1.

	T0	T1	*p*-Value
Weight (SDS)	2.13 ± 1.40	2.01 ± 1.56	0.25
BMI (SDS)	3.12 ± 1.17	3.03 ± 1.20	0.18
Weight gain (kg/month)	0.84 ± 0.55	0.29 ± 0.79	0.03 *
Insulin (µU/mL)	19.28 ± 12.29	18.16 ± 14.44	0.69
HOMA index	3.52 ± 2.18	3.33 ± 2.93	0.73
Triglycerides (mg/dL)	108.6 ± 51.0	110.5 ± 72.5	0.79
Total cholesterol (mg/dL)	191.4 ± 45.4	189.8 ± 29.4	0.98
HDL cholesterol (mg/dL)	56.8 ± 22.0	58.7 ± 25.2	0.46

* Statistically significant (*p*-value < 0.05).

**Table 4 children-08-00531-t004:** Main data at T0-T1-T2 of the 4 subjects treated with metformin from T1.

Subject	Weight (SDS)			BMI (SDS)			Weight Gain (kg/Month)			HOMA Index		
	T0	T1	T2	T0	T1	T2	T0	T1	T2	T0	T1	T2
1	4.16	4.05	3.95	3.43	3.33	3.28	1.02	−0.2	−0.16	6.16	2.06	1.4
3	3.72	3.93	3.82	3.23	3.32	3.26	1.5	1.6	−0.05	3.16	2.84	1.91
6	3.71	3.54	3.31	3.76	3.67	3.56	0.34	−0.34	−0.38	6.4	4.13	3.82
10	3.18	3.39	3.04	3.68	3.74	3.45	0.44	1.54	−0.87	4.7	3.45	3.74

## Data Availability

All records of patients are available in our local Informatic Database Trak-Care System and Local Piedmont Registry of Rare Diseases http://www.sistemapiemonte.it/cms/pa/sanita/servizi/251-registro-pazienti-affetti-da-malattie-rare (accessed on 1 March 2021).

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
