# Peer review of "Multidisciplinary Approach for Hypothalamic Obesity in Children and Adolescents: A Preliminary Study"

_children, 2021, doi:10.3390/children8070531_

Round 1
Reviewer 1 Report
see comments included in the pdf

Author Response
In the abstract we have changed the first sentence with a better definition of HO and corrected the sentence on the treatment effect.
In the introduction we have added the citation for refractoriness to lifestyle changes programs and we have deepened, thanks to the article you mentioned, the topic of bariatric surgery.
In the results we have also expressed, in the table of the caloric intake of the diets, the basal energy expenditure because the personalized diet (as we better explain in the new version of methods) was prepared considering the BEE, which was the only objective and calculated parameter, in addition to the dietary anamnesis and the physical activity, not considering strictly the recommended energy intake (the % of kcal of diets based only on recommended intake does not give a precise idea of the real dietitian’s work). We have combined table 2 and 4, as requested.
Finally, we have made the other minor corrections and clarifications requested.
Reviewer 2 Report
A multi-disciplinary approach for hypothalamic obesity in children and adolescents
Overall
The manuscript shares the experience of multidisciplinary approach to treat a small group of children and adolescents with hypothalamic obesity.
It also shares the 6-months outcomes of 4 participants that were treated with metformin.
This preliminary study has its importance to be published however major changes are needed.
The importance to publish this manuscript, even considering its important limitations is to 1) share more details on the approach used; 2) do not overstate their results and, 3) better address the existing limitations.
Suggestions
Title – the title should be modified to clearly address that it is a preliminary study. It is a very small sample size.
Abstract – Since the abstract it is important to make the correct reference to the variables and be fair to the results found. The weight and BMI are in fact SDS (abbreviation that should be explained when first presented to the readers) and there is no chance from T0 to T1.
For the 4 participants that received metformin – it is also weight and BMI SDS.
Introduction – There is a lack of rationale for the study. At the end of the introduction, it seems that the best approach to use for the population being study is to do bariatric surgery. The Introduction should be reviewed and a better rationale for the study proposed be presented. Even with the limitations previously referred regarding treatments based on lifestyle interventions for the patients with hypothalamic obesity, why the authors believe that their proposal is important and would benefit the patients?
Methods – Better description on the multidisciplinary treatment received is needed. How many visits, for how long, which professionals were involved and interacting with the patient, or the family? Were patients seen alone? Who answered and provided the information regarding dietary recalls? How many dietary recalls were collected? In what time frame?
Regarding the informed consent - was it obtained from the parents and was a informed assent obtained from the children (age < 18 years old)?
Statistical analysis – Why paired t-test was used to compare different groups of subjects at the same time-point?
Since sex difference was detected in the results, it would be interest to adjust the analysis for sex. Additionally, since the participants age range is very large, adjustment for age in the analyses is also recommended.
Results - On the addition to the information provided that the participants were receiving their usual doses of mediation for their endocrine comorbidities, were they under a good control? Also, it should be reported what was the hormone replacement therapy being received (GH, etc…)
For table 1 – height SDS should also be included since this is a pediatric population
How adherence to the multidisciplinary treatment was monitored? How adherence to metformin was checked on the 4 subjects?
Weight SDS and BMI SDS results should be reported as unchanged.
Discussion – More details should be given on the approach used and what were the new/innovative approaches that the authors hypothesize are being responsible for the relative short-term success.
In the discussion, Line 322-325, it should reported that the Weight SDS and BMI SDS was stable – unchanged. Do not overstate your findings, specially with a small sample size and the lack of control groups.
The finding in males vs. females is not a trend, P=0.04 as reported, it should be discussed as so. What is the authors interpretation of that finding?
Study limitations need to be better addressed in the discussion – the authors just present the limitation regarding the low number of participants receiving metformin. What about the study overall? The study has a small sample, large age range, lack of control groups, short follow-up (just 6 mo), potential interference of other medications being taken by the participants, etc.
Conclusion- it needs to be reviewed to reflect the findings – there were no changes in the BMI in the group investigated. Additionally, the conclusion looks like repetition of what was presented in the discussion.
Author Response
In the present version of our article, we have added more details on the approach used, we have corrected some parts to not overestimate the results and we have better explained the limitations.
The title has been modified to clarify that this is a preliminary study.
In the abstract we have specified that weight and BMI were expressed as SDS.
In the introduction we have deepened the rationale for the study, the differences from our approach to the classical intervention on lifestyle and we have corrected the part on bariatric surgery, explaining the limitations of this approach.
In the methods we have better described the characteristics of the multidisciplinary approach with the timing and modalities of visits and monitoring and the specialist involved.
In the new version of the statistical analysis (that we revised also with another program) we have specified that the paired t-test was used to compare only the same group at different time points, while the independent t-test was used to evaluate different groups of subjects at the same time-points.
Multivariate test analysis and Wilks' lambda did not indicate sex and age interferences on the main results, which consequently do not need to be corrected.
In the results more information on hormone replacement therapies have been added.
In table 1 we have included height SDS.
In the discussion we have added more details on the multidisciplinary approach, corrected the part where the results appeared overestimated and we have better addressed the limitations of the study, particularly regarding the low number of participants.
We have reviewed the conclusions, as required.
Finally, we have made the other minor corrections and clarifications requested.
Round 2
Reviewer 2 Report
The authors addressed several points brought up by the reviewers. Minor adjustments are still needed:
The report of results should be consistent:
- adjust the number of decimal places - line 260 and line 276.
- Change from commas to points as decimal separators - lines 305 and 306
Please, report the statistics for the multivariate test analysis that is being presented in lines 309-311.
Author Response
We revised the statistic analysis to better evaluate the effect of sex and age on the overall results and we have made the other minor corrections and clarifications requested.